# Potential Involvement of LncRNAs in Cardiometabolic Diseases

**DOI:** 10.3390/genes14010213

**Published:** 2023-01-13

**Authors:** Mirolyuba Ilieva, Shizuka Uchida

**Affiliations:** Center for RNA Medicine, Department of Clinical Medicine, Aalborg University, DK-2450 Copenhagen SV, Denmark or

**Keywords:** cardiovascular, genetics, lncRNA, metabolism

## Abstract

Characterized by cardiovascular disease and diabetes, cardiometabolic diseases are a major cause of mortality around the world. As such, there is an urgent need to understand the pathogenesis of cardiometabolic diseases. Increasing evidence suggests that most of the mammalian genome are transcribed as RNA, but only a few percent of them encode for proteins. All of the RNAs that do not encode for proteins are collectively called non-protein-coding RNAs (ncRNAs). Among these ncRNAs, long ncRNAs (lncRNAs) are considered as missing keys to understand the pathogeneses of various diseases, including cardiometabolic diseases. Given the increased interest in lncRNAs, in this study, we will summarize the latest trend in the lncRNA research from the perspective of cardiometabolism and disease by focusing on the major risk factors of cardiometabolic diseases: obesity, cholesterol, diabetes, and hypertension. Because genetic inheritance is unavoidable in cardiometabolic diseases, we paid special attention to the genetic factors of lncRNAs that may influence cardiometabolic diseases.

## 1. Introduction

Cardiometabolic diseases (CMD) account for 31% of all global deaths [1]. These life-long and life-threatening diseases are associated with an unhealthy lifestyle (e.g., obesity, physical inactivity, smoking), leading to diabetes mellitus and cardiovascular disease (CVD) [2], and more than 56,000 Danes are diagnosed with CVD annually [3]. In CMD (e.g., arrhythmic disorders, cardiomyopathies, Fabry disease, familial hypercholesterolemia, thoracic aortic aneurysms and dissections [4]), the genetic inheritance is unavoidable. For example, siblings of patients with cardiovascular disease (CVD) have a 40% increase of risk for CVD [5]. Thus, significant efforts have been spent to understand the genetic causes of CMD by genome sequencing the patients suffering from CMD. Through the usages of linkage disequilibrium (LD) and genome-wide association studies (GWAS), a number of loci are found to be associated with CMD [6,7,8,9,10]. Yet, only few are directly in the exons of protein-coding genes, leaving behind the large number of CMD-suspected loci to be unexplained.

The term junk DNA is long gone, and we now know that most of the mammalian genome are transcribed as RNA. Yet only few percent of them correspond to the exons of protein-coding genes, leaving behind the large number of RNAs as non-protein-coding (ncRNAs). These ncRNAs include ribosomal RNAs (rRNAs), transfer RNAs (tRNAs), short ncRNAs (e.g., small nuclear RNAs (snRNAs), small nucleolar RNAs (snoRNAs)), microRNAs (miRNAs), and other longer ncRNAs [11]. Long non-coding RNAs (lncRNAs) are any ncRNAs longer than 200 nucleotides (nt). Because the protein-centered research has not been able to fully elucidate the pathogeneses of CMD nor offer an ultimate cure for CMD, there are growing interests to study lncRNAs in the context of cardiovascular metabolism and CMD. Because lncRNAs occupy a large part of the human genome in comparison to protein-coding genes, increasing evidence suggests that disease-suspected loci based on LD and GWAS are in the loci with lncRNAs [12,13], including the well-known example of the lncRNA, *ANRIL* [official gene symbol: *CDKN2B-AS1* (CDKN2B antisense RNA 1)], which corresponds to the Chr9p21 risk locus for coronary artery disease based on GWAS studies [14,15,16,17,18,19,20,21]. Due to the decreased costs of performing both whole genome and transcriptome sequencing (i.e., DNA sequencing (DNA-seq) and RNA-seq, respectively), many lncRNAs have been identified in the disease-suspected loci, which calls for further investigation on how the mutations in lncRNAs themselves affect the susceptibility of each individual to a particular genetic disorder.

Here, we summarize the recent findings of lncRNAs in cardiovascular metabolism and CMD. Because there are many review articles summarizing lncRNAs in CVD, we will focus our attention on the lncRNAs that are involved in risk factors of CMD; namely, obesity, cholesterol, diabetes, and hypertension. To cope with the growing interest to study genetic causes of CMD, we will pay special attention to lncRNAs that lay within the CMD-suspected loci.

## 2. Obesity-Associated lncRNAs

Due to the increased consumption of high energy and calorie food (e.g., fat, oil, and sugar) coupled with physical inactivity, the number of obese people has increased dramatically worldwide in recent years [22,23,24]. Because obesity is an underlying cause and risk factor for many diseases (e.g., type II diabetes, CVD), the interest to study obesity-associated lnRNAs has increased recently. For example, the expression of one of the most well-studied lncRNA, H19 imprinted maternally expressed transcript (*H19*), is reduced in adipocytes of obese individuals [25]. The functional role of *H19* in CMD is well described, which includes negatively regulating hypertrophy [26,27,28] and pyroptosis (a form of cell death initiated by inflammation) [29] in cardiomyocytes. The reduced expression of *H19* inversely correlates with body mass index (BMI) in humans. Interestingly, *H19* overexpressing mice showed a protective effect against diet-induced obesity and improved insulin sensitivity in brown but not in white adipocytes (Figure 1A). Because BMI is a measure of body fat based on the height and weight of a person, it is used as an indicator of obesity. As such, GWAS and LD studies have been performed to identify BMI-associated loci [30,31]. Not surprisingly, several lncRNAs are shown to be in the BMI-associated loci, such as *LncOb rs10487505* variant between BMI and leptin level in pediatric non-alcoholic fatty liver disease [32], lncRNA transcript of the transcription factor 7 like 2 (*lncRNA-TCF7L2*) between BMI and biopolar disorder [33], MCHR2 antisense RNA 1 (*MCHR2-AS1*) between BMI and psychiatric patients [34], and the hypermethylation of the promoter of the lncRNA, TSPOAP1, SUPT4H1and RNF43 antisense RNA 1 (*TSPOAP1-AS1*), associated with obesity in overweight and obese Korean individuals [35]. These are just a few examples of BMI-associated lncRNAs reported to date, which most likely increase as more previously published GWAS data are re-analyzed from the standpoint of lncRNAs.

Obesity is characterized by the increased number of white adipocytes and adipocyte gene expression [36,37]. For example, the whole-body knockout mice of the lncRNA steroid receptor RNA activator 1 (*Sra1*) are resistant to high fat diet-induced obesity, with decreased epididymal and subcutaneous white adipose tissue (WAT) mass, and increased lean content (i.e., decreased percent of body fat) [38]. Mechanistically, it was shown previously that *Sra1* functions as a transcriptional coactivator of peroxisome proliferator-activated receptor gamma (Pparg), which is a master regulator of adipogenesis [39] (Figure 1B). It should be noted that *SRA1* encodes for both lncRNA and protein-coding transcripts (https://www.ensembl.org/Homo_sapiens/Gene/Summary?db=core;g=ENSG00000213523;r=5:140537340-140557677; accessed on 11 January 2023). Another example is adipocyte associated pyruvate carboxylase interacting lncRNA (*ADIPINT*), whose expression is increased in obesity and linked to fat cell size, adipose insulin resistance, and pyruvate carboxylase activity [40]. Mechanistically, *ADIPINT* binds to pyruvate carboxylase, which is a mitochondrial enzyme that catalyzes the ATP-dependent carboxylation of pyruvate to oxaloacetate in order to regulate lipid metabolism (Figure 1C). As adipocytes can be easily isolated and cultured, there are several screening studies to find lncRNAs associated with adipogenesis identifying hundreds of differentially expressed lncRNAs [41,42,43,44,45,46]. Thus, further functional and mechanistic studies are needed to elucidate the functional importance of these adipose-related lncRNAs.

As mentioned above, there are two types of adipocytes: brown and white adipocytes. While white adipocytes store excessive energy as trigycerides, brown adipocytes have significantly more mitochondria than white adipocytes and burn energy to generate heat [47]. Because of the benefits of having more brown adipocytes over white ones for reducing obesity, the interest to study brown adipose tissue (BAT) has increased in recent years [48]. For example, another well-studied lncRNA, X inactive specific transcript (*XIST*), regulates brown preadipocytes differentiation by directly binding to CCAAT enhancer binding protein alpha (CEBPA, also known as C/EBPα) [49]. Besides BAT, browning of WAT (also called as adipose tissue browning) is gaining attention as a potential therapeutic target to induce thermogenically active adipocytes in WAT to increase energy expenditure and counteract weight gain [50]. During adipose tissue browning, several lncRNAs have been identified and functionally studied, including CCCTC-binding factor (zinc finger protein)-like, opposite strand (*Ctcflos*) [51], FOXC2 antisense RNA 1 (*FOXC2-AS1*) [52], and predicted gene 13133 (*Gm13133*) [53] (Figure 1D). Thus, lncRNAs might hold keys to reduce obesity, thereby reducing CMD.

## 3. Cholesterol-Associated lncRNAs

Although cholesterol is needed for our body, an excessive amount of cholesterol in blood leads to coronary artery disease (CAD, also known as atherosclerosis, a type of cardiovascular disease), which is caused by the blockade of arteries by cholesterol and other substances that form atherosclerotic plaques [54]. Because CAD is the third leading cause of mortality worldwide [55], there is a high interest to understand the pathogenesis and potentially find cures for CAD. Besides the most well-known CAD-associated lncRNA, *ANRIL*, as mentioned in the Introduction section, there are several lncRNAs associated with CAD [56,57]. For example, the expression of the lncRNA FXYD6 antisense RNA 1 (*FXYD6-AS1*, also known as *RP11-728F11.4*), is increased in atherosclerosis plaques [58] (Figure 2A). In the cultured monocyte-derived macrophages (THP-1 cells), lentivirus-induced overexpression of *FXYD6-AS1* resulted in higher sodium–potassium adenosine triphosphatase (Na^+^/K^+^-ATPase) activity, intracellular cholesterol accumulation, and increased proinflammatory cytokine production. These phenotypes were also observed in apolipoprotein E deficient (apoE)−/−mice on a C57BL/6J genetic background fed with a Western diet (a defined mouse model of atherosclerosis as well as non-alcoholic steatohepatitis [59,60]), where overexpression of *FXYD6-AS1* increased proinflammatory cytokine production and augmented atherosclerotic lesions. Mechanistically, *FXYD6-AS1* binds EWS RNA-binding protein 1 (EWSR1), which binds to the promoter region of FXYD domain-containing ion transport regulator 6 (*FXYD6*), the modulator of Na^+^/K^+^-ATPase. Using the similar in vitro and in vivo experimental systems, another well-known lncRNA, KCNQ1 opposite strand/antisense transcript 1 (*KCNQ1OT1*), was also shown to be involved in CAD. *KCNQ1OT1* functions as a microRNA (miRNA) sponge to sequester *miR-452-3p*, which binds to histone deacetylase 3 (*HDAC3*) and thereby inhibits the expression of ATP binding cassette subfamily A member 1 (*ABCA1*) (Figure 2B). This coordinated pathway in macrophages results in lipid accumulation and accelerates the atherosclerosis development [61].

As exemplified by *ANRIL*, there are several polymorphisms in lncRNA genes associated with CAD [62,63], including growth arrest specific 5 (*GAS5*) [64], *H19* [65,66,67], *HOX* transcript antisense RNA (*HOTAIR*) [68], metastasis-associated lung adenocarcinoma transcript 1 (*MALAT1*) [69,70,71], and tumor protein p53 pathway corepressor 1 (*TP53COR1*, also known as *lincRNA-p21*) [72]. According to the latest annotation provided by the Ensembl database [73] (https://www.ensembl.org/Homo_sapiens/Gene/Summary?db=core;g=ENSG00000251562;r=11:65497688-65506516; accessed on 11 January 2023), there are 17 transcripts (isoforms) of *MALAT1*. The length of the longest transcript is 8762 nucleotides (nt). As such, a simple search of the NCBI database of genetic variation, dbSNP database [74] (https://www.ncbi.nlm.nih.gov/snp/?term=malat1; accessed on 11 January 2023), indicates that there are 7341 records (dbSNP RefSNP IDs), suggesting that there are many sequence variants within *MALAT1* because it is a long transcript. When the term *MALAT1* was searched via the ClinVar database [75] (freely available, public archive of human genetic variants and interpretations of their significance to disease) (https://www.ncbi.nlm.nih.gov/clinvar/?term=malat1%5Bgene%5D&redir=gene; accessed on 11 January 2023), 14 records (e.g., ependymoma and glycogen storage disease, type V, Bardet-Biedl syndrome) were found, further suggesting that the lncRNA polymorphisms are associated with various diseases, not only CAD. Not surprisingly, the longer the lncRNA, the more genetic variations are found, which increases the chance of finding associated diseases. To this end, re-analysis of published GWAS studies and their data sets is important to further identify lncRNAs in disease-suspected loci.

## 4. Diabetes-Associated lncRNAs

Type II diabetes (T2D) is a chronic disease condition with high blood glucose level caused by the inability of the pancreas to produce enough insulin to regulate glucose level [76,77]. The prevalence of T2D is high, affecting approximately 6.28% of the world’s population [78]. The causes of T2D include obesity, physical inactivity, and genetic factors [79,80,81]. If left untreated, T2D can cause other health problems, including CVD [82], high blood pressure (hypertension) [83], vision loss (diabetic retinopathy) [84], and kidney disease [85]. Thus, intensive T2D research is ongoing to understand the impact of T2D to other health problems, including CMD. Not surprisingly, several lncRNAs are implicated to be involved in the pathogenesis of T2D [86,87,88,89]. For example, the lncRNA, antisense Igf2r RNA (*Airn*, also known as *Air*), is a paternally imprinted gene located in antisense orientation to the imprinted, but maternally derived, insulin-like growth factor 2 receptor (*Igf2r*) protein-coding gene (Figure 3A). We previously showed that *Airn* binds to the RNA-binding protein, insulin-like growth factor 2 mRNA binding protein 2 (Igf2bp2, also known as IMP2), and controls the translation of messenger RNAs (mRNAs) (e.g., biglycan, inhibin β-A) to regulate the cardiomyocyte resistance to stress [90]. Recently, another group reported that *Airn* alleviates diabetic cardiac fibrosis by regulating degradation of transformation-related protein 53 (*Trp53*, also known as *p53*) mRNA in N^6^-methyladenosine (m^6^A) RNA methylation manner because Igf2bp2 is a m^6^A reader [91]. This study and others clearly link the importance of lncRNAs in pathogenesis of T2D. Furthermore, lncRNAs can be the determinants to understand how RNA metabolisms are regulated through RNA modifications (epitranscriptomics), as we recently reviewed [92].

Diabetic cardiomyopathy (DCM) is a type of CVD caused by diabetes, resulting in contractile dysfunction and leading to heart failure [93]. Several lncRNAs have been identified to be involved in the pathogenesis of DCM (Figure 3B). For example, the lncRNA DCM-related factor (*DCRF*) functions as a miRNA sponge to sequester *miR-551b-5p*, which targets protocadherin 17 (*Pcdh17*) to regulate autophagy in rat cardiomyocytes [94]. Another example is the lncRNA taurine upregulated gene 1 (*Tug1*). It is up-regulated in the DCM mouse model based on the intraperitoneally injection of streptozotocin (an antibiotic that produces pancreatic islet β-cell destruction [95]), while down-regulation of *Tug1* improved cardiac function and myocardial fibrosis in the DCM mice by decreasing the levels of extracellular matrices (e.g., collagens). Mechanistically, *Tug1* binds *miR-145a-5p*, which targets cofilin 2, muscle (*Cfl2*) [96]. Besides these two lncRNAs, there are other lncRNAs reported to function as miRNA sponges in DCM [97,98,99,100], suggesting a popular mechanism of action of lncRNAs to be investigated. In this regard, several screening studies to identify lncRNAs functioning as miRNA sponges in DCM have been conducted [101,102,103]. However, it should be noted that many lncRNAs are lowly expressed compared to mRNAs and miRNAs. Thus, it is advised that researchers investigating lncRNAs as miRNA sponges to report the copy numbers of lncRNAs and their target miRNAs to be sure that it is possible for the target lncRNA to be abundantly present in a target cell compared to the available miRNA in the corresponding cell to sequester the target miRNA, which gives rise to the phenotypes that are observed in their experimental systems.

## 5. Hypertension-Associated lncRNAs

Hypertension (high blood pressure) is a condition in which blood pressure is higher than normal. According to the current guideline by the American College of Cardiology/American Heart Association, hypertension is defined as a cut point of systolic blood pressure (SBP) ≥130 mm Hg and/or diastolic blood pressure (DBP) ≥80 mm Hg [104]. As the risk of hypertension increases due to the underlying conditions (e.g., diabetes and obesity), it is no surprise that several lncRNAs are identified to be linked to hypertension. For example, angiotensin II (AngII)-induced lncRNA *Alivec* [Angiotensin II-induced lncRNA in vascular smooth muscle cells (VSMCs) eliciting chondrogenic phenotype] is shown to be induced by the growth factors AngII and platelet-derived growth factor (PDGF) and the inflammatory cytokine tumor necrosis factor α (TNF-α) via the action of the transcription factor SRY-box transcription factor 9 (Sox9) in rat VSMCs [105] (Figure 4A). Mechanistically, *Alivec* binds tropomyosin 3 (Tpm3) and heterogeneous nuclear ribonucleoprotein A2/B1 (Hnrnpa2b1) to regulate chondrogenic transformation of VSMCs implicated in vascular dysfunction and hypertension.

Pulmonary hypertension is caused by high pressure in the blood vessels from the heart to the lungs, which affects both the lungs and the heart [106,107]. It is characterized by pathological changes to signaling pathways (e.g., endothelin, nitric oxide, prostacyclin, transforming growth factor β (TGFB) signaling pathways) [108,109], and a number of lncRNAs have been identified to play key roles in these signaling pathways [110,111,112,113]. Besides binding to miRNAs and RNA-binding proteins, lncRNAs can interact with DNA to regulate the binding of epigenetic and/or transcription factors to the genome, thereby, regulating epitranscriptomic and/or transcriptomic changes (Figure 4B). Increasing evidence suggests that lncRNAs interact with DNA by forming triplex structures [114,115]. Recently, an interesting study was published. In this study, the authors conducted a functional screening of DNA:DNA:RNA triplex-forming lncRNA in human endothelial cells to identify the lncRNA HIF1A antisense RNA 1 (*HIF1A-AS1*) [116]. HIF1A-AS1 is down-regulated in patients suffering from pulmonary hypertension. Mechanistically, *HIF1A-AS1* acts as an adapter for the repressive human silencing hub complex (HUSH). Originally identified as an essential lncRNA gene for the normal development of the heart and body wall in mice [117], FOXF1 adjacent non-coding developmental regulatory RNA (*FENDRR*) is recently shown to act in a similar mechanism as *HIF1A-AS1* by forming triplex structure in the promoter region of dynamin 1 like (*DNM1L*, also known as *DRP1*) [118]. These two studies exemplified how the field of lncRNAs has moved from simply identifying the binding partners (e.g., DNA, RNA, and proteins) to detailed mechanistic studies. Although it has not been discussed in these original studies, an open question is whether in such genomic regions that lncRNAs show any disease-specific or -associated mutations or not. It is speculated that bioinformatic analysis by combining genomic mutation data and the binding sites of lncRNAs might reveal further involvement of lncRNAs in disease pathogeneses, including pulmonary hypertension.

## 6. Conclusions

As summarized above, lncRNAs can be the key to unravel the pathogeneses of CMD, which may lead to the potential cures for CMD [119]. As increasing evidence suggests that genetic factors of CMD are important, understanding how single nucleotide polymorphisms (SNPs) in lncRNAs are linked to various diseases, including CMD, is urgently needed. For this purpose, the readers are advised to explore lncRNA polymorphisms associated with diseases through the NHGRI-EBI Catalog of human genome-wide association studies (https://www.ebi.ac.uk/gwas/search?query=BMI) (accessed on 5 December 2022). Although there are several clinical trials underway to investigate the correlation of lncRNA expression to different diseases as potential diagnostic biomarkers [120,121,122,123], the therapeutic approach to use lncRNAs is still not in any clinical trials [119]. Thus, the further mechanistic investigation of each lncRNA is necessary to advance the lncRNA therapeutics in near future.

## Figures and Tables

**Figure 1 genes-14-00213-f001:**
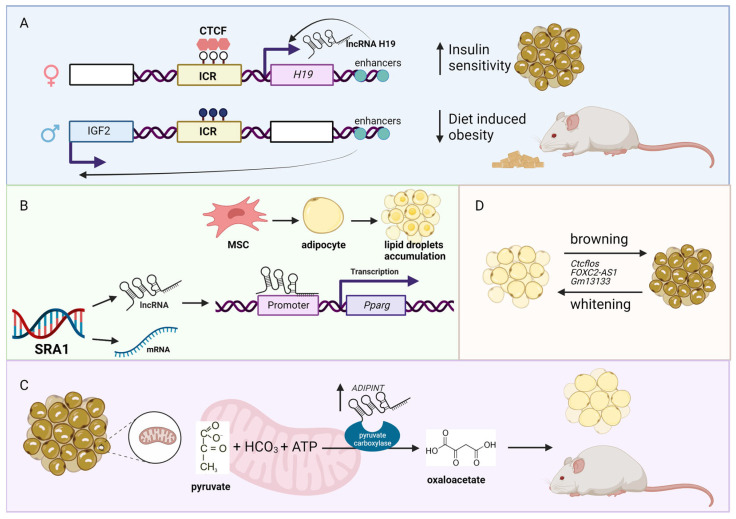
LncRNAs associated with obesity. (**A**) The *H19*–*IGF2* locus consists of a maternal unmethylated (white lollipops in the image above) imprinting control region (ICR; yellow box). ICR is bound by the insulator protein CCCTC-binding factor (CTCF), which prevents access of the *IGF2* promoter (blue box represents the exon from the paternal chromosome) to downstream enhancers. The overexpression of the lncRNA *H19* (pink box represents the exon from the maternal chromosome) in mice has a protective effect against diet-induced obesity and increased insulin sensitivity. In the image above, white boxes represent the repressed genes, while black lollipops represent the methylated CpG sites. (**B**) The *SRA1* gene encodes for protein-coding mRNA and lncRNA *Sra1*. The lncRNA *Sra1* functions as a transcription co-activator of peroxisome proliferator-activated receptor gamma (Pparg), which is a regulator of adipocytes differentiation and lipid accumulation. (**C**) The lncRNA *ADIPINT* is upregulated in obesity. This lncRNA is responsible for adipocyte size, insulin resistance, and pyruvate carboxylase activity. (**D**) Several lncRNAs have been identified to be involved in the process of browning and whitening in the adipose tissue, which could serve as potential therapeutic targets. Figure created with BioRender.com (accessed on 10 January 2022).

**Figure 2 genes-14-00213-f002:**
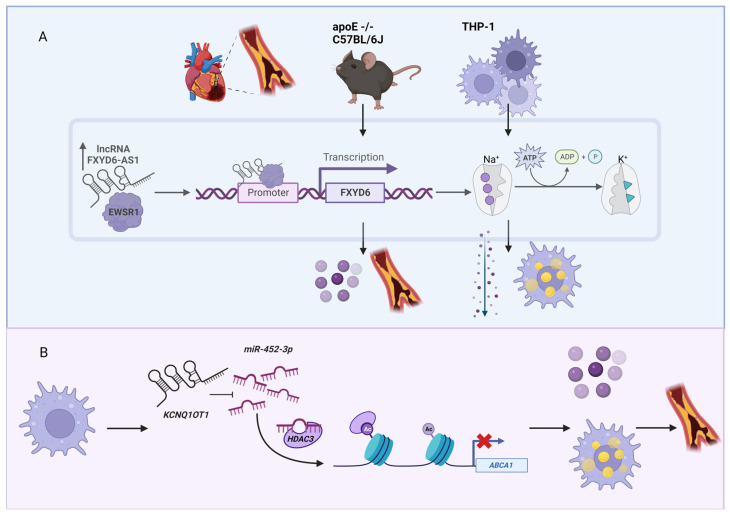
LncRNAs associated with atherosclerosis development. (**A**) The lncRNA *FXYD6-AS1* is overexpressed in the patients with atherosclerosis as well as in the apoE deficient mice fed with a Western diet. Together with EWS RNA binding protein 1 (EWSR1), *FXYD6-AS1* regulates the expression of the protein-coding gene *FXYD6*, which is the main modulator of Na^+^/K^+^-ATPase activity. (**B**) The lncRNA *KCNQ1OT1* functions as a miRNA sponge. This lncRNA sequesters *miR-452-3p*, which binds to histone deacetylase 3 (HDAC3) and thus inhibits the expression of ATP binding cassette subfamily A member 1 (*ABCA1*). Phenotypically, this cascade results in lipid accumulation and proinflammatory cytokine release in macrophages and accelerates the atherosclerotic development. Figure created with BioRender.com (accessed on 9 December 2022).

**Figure 3 genes-14-00213-f003:**
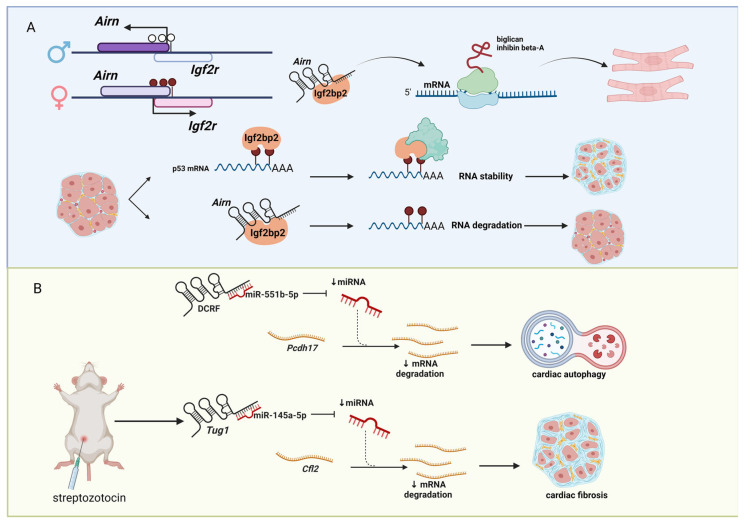
LncRNAs associated with diabetes. (**A**) The lncRNA *Airn* is paternally imprinted and located in antisense orientation to the maternally imprinted derived protein-coding gene *Igf2r*. *Airn* binds to the RNA-binding protein Igf2bp2 and controls the translation of mRNAs (e.g., biglycan and inhibin β-A) to modulate the resistance to stress in cardiomyocytes. Similarly, in diabetic cardiac fibrosis, *Airn* binds to Igf2bp2 to regulate the degradation of *Trp53* mRNA in a m^6^A manner. (**B**) Several lncRNAs are involved in diabetic cardiomyopathy (DCM) by functioning as miRNA sponges. The lncRNA *DCRF* sequester *miR551b-5p* targets protocadherin 17 mRNA to regulate autophagy in rat cardiomyocytes. The upregulation of the lncRNA *Tug1* in the DCM streptozotocin-based mouse model leads to an increasing level of extracellular matrix and myocardial fibrosis development. Figure created with BioRender.com (accessed on 9 December 2022).

**Figure 4 genes-14-00213-f004:**
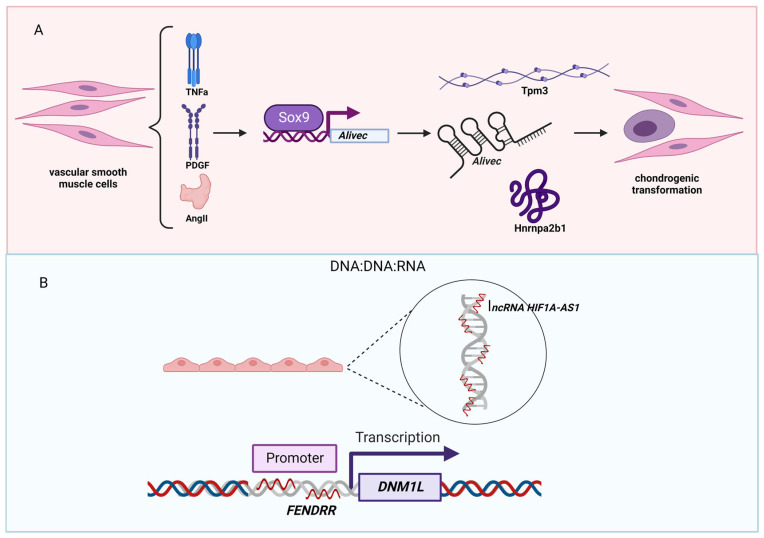
LncRNAs associated with hypertension. (**A**) In vascular smooth muscle cells, the expression of the lncRNA *Alivec* is induced by the growth factors AngII, TNFa, and PDGF via the transcription factor Sox9. Mechanistically, *Alivec* binds to tropomyosin 3 and Hnrnpa2b1 and regulates chondrogenic transformation of smooth muscles observed in vascular dysfunction and hypertension. (**B**) A mechanism by which lncRNAs implement a regulatory function through the DNA:DNA:RNA triplex formation. In human endothelial cells, the lncRNA *HIF1A-AS1* serves as an adapter for the repressive human silencing hub complex (HUSH). Another lncRNA, *FENDRR*, forms a triplex structure in the promotor region of dynamin 1 like (*DNM1L*). Figure created with BioRender.com (accessed on 9 December 2022).

## Data Availability

Not applicable.

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
