# Peer review of "Potential Involvement of LncRNAs in Cardiometabolic Diseases"

_genes, 2023, doi:10.3390/genes14010213_

Round 1

Reviewer 1 Report

In this manuscript, the authors summarized the latest trend in the lncRNA research in cardiometabolic diseases (CMD) by focusing on the major risk factors: obesity, cholesterol, diabetes, and hypertension. They clearly and interestingly introduced several lncRNAs associated with CMD, e.g., H19, Sra1, RP11-728F11.4, Airn and Alivec. This manuscript is well-organized and easy to understand. I have no major concerns. A few minor concerns are listed below:

1.       Lines 24 and 25, “These life-long and -threatening diseases are caused by unhealthy lifestyle”. Incorrect statement. Not all CMDs are caused by an unhealthy lifestyle.

2.       Lines 68 and 69, reference “27” and “28”, which refer to bipolar disorder and psychiatric patients, respectively, are not well associated with the topic of this review. Please include references that are more related to CMD.

3.       For Figure 1 (A), more information about the figure legend is needed to understand it fully (e.g., the meaning of rectangles with different colors, male symbol, female symbol), and please indicate the references for Figure 1 (A).

4.       lncRNA H19 is well studied in CMD; please add a brief summary of the mechanism and its roles in CMD.

5.       For Figure 3 (A), no illustration of the regulation of Airn to Trp53 in the figure legend.

Author Response

In this manuscript, the authors summarized the latest trend in the lncRNA research in cardiometabolic diseases (CMD) by focusing on the major risk factors: obesity, cholesterol, diabetes, and hypertension. They clearly and interestingly introduced several lncRNAs associated with CMD, e.g., H19, Sra1, RP11-728F11.4, Airn and Alivec. This manuscript is well-organized and easy to understand. I have no major concerns. A few minor concerns are listed below:

  1. Lines 24 and 25, “These life-long and -threatening diseases are caused by unhealthy lifestyle”. Incorrect statement. Not all CMDs are caused by an unhealthy lifestyle.

Response: Thank you very much for your praise. The above sentence was modified as follow:

“These life-long and -threatening diseases are associated with unhealthy lifestyle (e.g., obesity, physical inactivity), leading to diabetes mellitus and cardiovascular disease (CVD) [2], which more than 56,000 Danes are diagnosed with CVD annually [3].”

  1. Lines 68 and 69, reference “27” and “28”, which refer to bipolar disorder and psychiatric patients, respectively, are not well associated with the topic of this review. Please include references that are more related to CMD.

Response: We extensively searched for lncRNAs associated with body mass index (BMI), yet we could only identify one another publication that is in the context of CMD, which is now added to the sentence as follow:

“Not surprisingly, several lncRNAs are shown to be in the BMI-associated loci, such as LncObrs10487505 variant between BMI and leptin level in pediatric non-alcoholic fatty liver disease [30], lncRNA transcript of the transcription factor 7 like 2 (lncRNA-TCF7L2) between BMI and biopolar disorder [31], MCHR2 antisense RNA 1 (MCHR2-AS1) be-tween BMI and psychiatric patients [32], the hypermethylation of the promoter of the lncRNA, TSPOAP1, SUPT4H1 and RNF43 antisense RNA 1 (TSPOAP1-AS1), associated with obesity in overweight and obese Korean individuals [33].”

  1. For Figure 1 (A), more information about the figure legend is needed to understand it fully (e.g., the meaning of rectangles with different colors, male symbol, female symbol), and please indicate the references for Figure 1 (A).

Response: The figure was modified and the corresponding legend parts were modified as follow:

“Figure 1. LncRNAs associated with obesity. (A) The H19–IGF2 locus consists of a maternal unmethylated (white lollipops in the image above) imprinting control region (ICR; yellow box). ICR is bound by the insulator protein CCCTC-binding factor (CTCF) that prevents access of the IGF2promoter (blue box represents the exon from the paternal chromosome) to downstream enhancers. The overexpression of the lncRNA H19 (pink box represents the exon from the maternal chromosome) in mice has a protective effect against diet induced obesity and increase insulin sensitivity. In the image above, white boxes rep-resent the repressed genes, while black lollipops represent the methylated CpG sites.”

  1. lncRNA H19 is well studied in CMD; please add a brief summary of the mechanism and its roles in CMD.

Response: The following sentence was added:

“The functional role of H19 in CMD is well described, which include negatively regulating hypertrophy [24-26] and pyroptosis (a form of cell death initiated by inflammation) [27] in cardiomyocytes.”

  1. For Figure 3 (A), no illustration of the regulation of Airn to Trp53 in the figure legend.

Response: The following sentence was added to the figure legend:

“Similarly, in diabetic cardiac fibrosis, Airn binds to Igf2bp2 to regulate the degradation of Trp53mRNA in a m6A manner.”

Reviewer 2 Report

The manuscript is interesting and well-organized. Most relevant topics were covered properly. References were properly cited in the text.

There are some suggestions to improve the quality of the manuscript and expand its scope, as suggested below:

1. The authors discussed the potential mechanistic involvement of lncRNAs in the pathogenesis of cardiometabolic disorders. Yet, the therapeutic applications of these lncRNAs were not discussed adequately. It is recommended to highlight some reported/potential therapeutic applications of them. 

2. The research on lncRNAs has grown recently. Nevertheless, the clinical translation of such a technology will be challenged by the poor delivery efficiency to the target cells in vivo. There are many successful approaches with other similar nucleic acid therapeutics that reached the market. The authors can refer to this important point in the future perspectives.

Author Response

The manuscript is interesting and well-organized. Most relevant topics were covered properly. References were properly cited in the text.

There are some suggestions to improve the quality of the manuscript and expand its scope, as suggested below:

  1. The authors discussed the potential mechanistic involvement of lncRNAs in the pathogenesis of cardiometabolic disorders. Yet, the therapeutic applications of these lncRNAs were not discussed adequately. It is recommended to highlight some reported/potential therapeutic applications of them.

  1. The research on lncRNAs has grown recently. Nevertheless, the clinical translation of such a technology will be challenged by the poor delivery efficiency to the target cells in vivo. There are many successful approaches with other similar nucleic acid therapeutics that reached the market. The authors can refer to this important point in the future perspectives.

Response: Thank you very much for your praise. The following sentences were added as closing statements in the Conclusion section:

"Although there are several clinical trials underway to investigate the correlation of lncRNA expression to different diseases as potential diagnostic biomarkers [116-119], the therapeutic approach to use lncRNAs is still not in a clinical trial [120]. Thus, the further mechanistic investigation of each lncRNA is necessary to advance the lncRNA thera-peutics in near future."
